# Chitosan-Coated PLGA Nanoparticles Encapsulating Triamcinolone Acetonide as a Potential Candidate for Sustained Ocular Drug Delivery

**DOI:** 10.3390/pharmaceutics13101590

**Published:** 2021-09-30

**Authors:** Madhuri Dandamudi, Peter McLoughlin, Gautam Behl, Sweta Rani, Lee Coffey, Anuj Chauhan, David Kent, Laurence Fitzhenry

**Affiliations:** 1Ocular Therapeutics Research Group, Pharmaceutical and Molecular Biotechnology Research Centre, Waterford Institute of Technology, X91 K0EK Waterford, Ireland; pmcloughlin@wit.ie (P.M.); gbehl@wit.ie (G.B.); srani@wit.ie (S.R.); LCOFFEY@wit.ie (L.C.); LFITZHENRY@wit.ie (L.F.); 2Department of Chemical and Biological Engineering, Colorado School of Mines, Colorado, CO 80401, USA; chauhan@mines.edu; 3The Vision Clinic, R95 XC98 Kilkenny, Ireland; dkent@liverpool.ac.uk

**Keywords:** nanoparticles, posterior segment eye diseases, PLGA, chitosan, chitosan-coated nanoparticles, triamcinolone acetonide, corticosteroid, ocular drug delivery

## Abstract

The current treatment for the acquired retinal vasculopathies involves lifelong repeated intravitreal injections of either anti-vascular endothelial growth factor (VEGF) therapy or modulation of inflammation with steroids. Consequently, any treatment modification that decreases this treatment burden for patients and doctors alike would be a welcome intervention. To that end, this research aims to develop a topically applied nanoparticulate system encapsulating a corticosteroid for extended drug release. Poly (lactic-co-glycolic acid) (PLGA) nanoparticles (NPs) supports the controlled release of the encapsulated drug, while surface modification of these NPs with chitosan might prolong the mucoadhesion ability leading to improved bioavailability of the drug. Triamcinolone acetonide (TA)-loaded chitosan-coated PLGA NPs were fabricated using the oil-in-water emulsion technique. The optimized surface-modified NPs obtained using Box-Behnken response surface statistical design were reproducible with a particle diameter of 334 ± 67.95 to 386 ± 15.14 nm and PDI between 0.09 and 0.15. These NPs encapsulated 55–57% of TA and displayed a controlled release of the drug reaching a plateau in 27 h. Fourier-transform infrared spectroscopic (FTIR) analysis demonstrated characteristic peaks for chitosan (C-H, CONH2 and C-O at 2935, 1631 and 1087 cm^−1^, respectively) in chitosan-coated PLGA NPs. This result data, coupled with positive zeta potential values (ranged between +26 and +33 mV), suggests the successful coating of chitosan onto PLGA NPs. Upon coating of the NPs, the thermal stability of the drug, polymer, surfactant and PLGA NPs have been enhanced. The characteristics of the surface-modified NPs supports their use as potential candidates for topical ocular drug delivery for acquired retinal vasculopathies.

## 1. Introduction

According to the World Health Organisation (WHO), the majority of people with vision loss and vision impairment are aged over 50 [1]. As the percentage of aging in the global population continues to increase, the number of people suffering from acquired ocular diseases will also significantly rise (~20% of Europe’s population are now over 65) [2]. The acquired retinal vasculopathies, including diabetic retinopathy (DR), venous occlusive disease (VOD) and age-related macular degeneration (AMD) are the most common causes of vision loss in the elderly [3]. All of these conditions are typically treated by regular intravitreal injections that are expensive and can be associated with serious side effects such as cataracts, retinal detachment and endophthalmitis [4]. Together with an increased financial burden, clinical level of anxiety and depression was seen in patients receiving intravitreal injections, which is hindering the patients’ quality of life [5]. As such, there is a compelling necessity to develop a patient friendly treatment with negligible side effects such as topical formulation or controlled release drug delivery system which could reduce or negate the need for current intravitreal eye injections. The main pathological factors contributing to AMD, DR, VOD and macular edema appear to be inflammation, retinal fluid accumulation with a higher vascular permeability [6]. Synthetic glucocorticosteroids (GCs) are commonly used drugs to treat a wide variety of inflammatory diseases in the eye [7]. They are mainly used for their immunosuppressive and anti-inflammatory properties, which are regulated through signal transduction by glucocorticoid receptors. Triamcinolone acetonide (TA), a member of the GCs family, has been shown to inhibit the secretion of VEGF, the production of cytokines and decrease vascular permeability [8,9].

The major issue with the use of hydrophobic drugs, including TA, is minimal aqueous solubility leading to suboptimal bioavailability. Loading hydrophobic drugs into nanoparticulate systems can help overcome this complication [10]. Developing a topical formulation is a significant challenge, as the eye is a complex organ with multiple anatomical, biochemical, and physiological barriers restricting the entry of drug molecules to the site of action. Since the initial use of nanoparticulate systems in medicine to date, they have revolutionized drug delivery and have also shown promising research outcomes for treating complex ocular diseases [11,12,13]. Different forms of nanomaterials being tested for anterior and posterior segment eye diseases include NPs [14], nano-micelles [15], liposomes [16], dendrimers [17], nanoparticle-loaded contact lenses [18] and sub-conjunctival implants with nanostructures [19], etc. Liposomes and micelles have limitations such as low stability, leakage of entrapped drug and premature release of the loaded drug [20,21]. The changes in NPs-loaded contact lenses during the storage period like burst release of the drug before administration, swelling and optical transparency, etc., still need to be addressed [22]. For elderly patients wearing contact lenses and inserting implants with minor surgery leads to poor patient compliance. Recently, polymeric NPs has been studied for topical ocular delivery and shown promising results [23,24,25]. Considering the size of NPs, they are likely to have greater diffusivity through biological membranes like the corneal epithelium compared to a suspension of hydrophobic drugs. Previous research on NPs for ocular drug delivery has demonstrated an increase in corneal permeability of the incorporated drugs when compared to their drug suspension [26,27,28]. Additionally, the higher surface area of nanoparticulate systems enables enhanced interaction with the epithelial layers of the eye leading to the increased retention time of topically administered drug delivery systems [29].

The biodegradable polymer, poly lactic-co-glycolic acid (PLGA), is approved by the United States food and drug administration (FDA) and European Medicines Agency (EMA) and proved its suitability for drug delivery [30]. Additionally, there is a commercial PLGA based formulation called Ozurdex^®^ to treat VOD, DR and posterior uveitis which highlights the ocular tolerance of PLGA [31]. PLGA NPs have shown good stability with prolonged release of the drug in many studies but they lack mucoadhesion [32]. This property enhances the ocular surface retention time, which plays a vital role when targeting the topical route for posterior segment diseases as the loss of NPs in tear drainage is the major limitation [33,34,35]. The surface modification of PLGA NPs with natural and biodegradable polymers like chitosan has proven to be beneficial for ocular drug delivery [23,36,37]. Polymers with mucoadhesive properties bind to mucin through various mechanisms such as hydrogen bonds, electrostatic interactions, polymer chain inter-diffusion and van der Waal forces [38]. These surface modified NPs may have the potential to increase the stability of the encapsulated drug, reducing the initial burst release of the drug and the in vivo targeting ligands may conjugate with free amine groups of chitosan [39,40]. Conversion of charge from negative to positive may enhance cellular adhesion and retention time of the formulation at the target site [41]. The adsorption of chitosan onto PLGA NPs follows a multilayer behavior with the reason for adsorption being found to be the cationic nature of chitosan and also the non-uniform porous surface of PLGA NPs [42]. Considering these parameters, the current research was focused on developing a topically applied nanoparticulate system exhibiting extended drug release for the treatment of posterior segment disease. This system consists of TA-loaded chitosan-coated PLGA NPs prepared by an emulsion technique. Emulsifier or stabilizer, such as Pluronic^®^ F-127 (PF-127), used in the nanoparticle preparation forms a steric barrier around the nanoparticle with both PEO (polyethylene oxide) and PPO (polypropylene oxide) moieties, thus keeping the NPs uniformly dispersed [43]. Optimization of the NPs was achieved using a response surface design of experiments. The prepared NPs were investigated for particle size, polydispersity index (PDI), zeta potential (ZP) and percent drug encapsulation efficiency (EE). The NPs were investigated for thermal stability, characterized by FT-IR, and examined for in vitro drug release to assess their suitability as drug carriers. As the research on nanoparticulate systems continues with ocular use, the scalability, safety, and robustness of these systems will be established leading to a new and highly efficient nanoparticulate system available for patient use.

## 2. Materials and Methods

### 2.1. Materials

Triamcinolone acetonide (TA) (MW: 434.50 g/mol with purity >99%) and water-soluble chitosan (MW: 10–100 KDa with a deacetylation degree >90%) were procured from Carbosynth Ltd., Berkshire, UK. PLGA (DL-lactide/Glycolide copolymer, ratio M/M%: 50/50, MW: 38,000–54,000), Pluronic^®^ F-127 (MW: ~12,600 g/mol), poly (vinyl alcohol) (MW: 89,000–98,000), dichloromethane (purity ≥99.8%), tween^®^ 80, sodium azide (purity >99.5%) and phosphate buffer saline tablets (0.01 M phosphate buffer), were purchased from Sigma Aldrich, Arklow, Ireland.

### 2.2. Methods

#### 2.2.1. Preparation and Optimization of Chitosan-Coated PLGA Nanoparticles

The NPs were prepared by a single emulsion technique which was a modification of a previously published double emulsion method [36]. PLGA (3.5 mg/mL) and TA (1 mg) were dissolved in 2 mL of dichloromethane (DCM) and this mixture was added dropwise to 10 mL of the aqueous solution containing stabilizer (0.25% (*w*/*v*) PVA or 1% (*w*/*v*) PF-127). Upon which, the mixture was sonicated for 10 min and left stirring overnight to evaporate the organic solvent. Finally, the nanosuspension was centrifuged at 3000 rpm for 15 min at 4 °C to remove any unentrapped drug. The supernatant obtained was subjected to high-speed centrifugation at 15,000 rpm for 30 min at 4 °C to recover the NPs. Chitosan-coated PLGA NPs were obtained by incubating PLGA NPs in chitosan solution (1:1 (*v*/*v*)) overnight by stirring at room temperature. After the incubation, the NPs were collected by centrifugation at 15,000 rpm for 30 min (NPs redispersed in Milli-Q^®^ water with pH 7). Considering the reproducibility of the NPs in the screening experiments, the formulation optimization experiments were designed using the design of experiments (DOE). For the optimization of NPs, a Box-Behnken, surface-response DOE was chosen. The three independent variables used were PLGA (3.5–4.5 mg/mL), PF-127 (1–2% (*w*/*v*)) and chitosan (1–2% (*w*/*v*)), while the four dependent response variables were particle size, PDI, ZP and %EE. This DOE generated 30 formulations (Appendix A), and the NPs with the same composition were given one formulation code starting from E1 to E13 for PLGA NPs (prepared with the same composition as Table 1 without chitosan) and CS-E1 to CS-E13 for chitosan-coated NPs as represented in Table 1.

The ideal concentrations of independent variables were selected based on the responses obtained. Additionally, two formulations were chosen using point prediction and the experimental values obtained were compared to predicted values to verify the experimental model chosen.

#### 2.2.2. Particle Size, Polydispersity Index and Zeta Potential

The particle size, PDI and ZP were determined using dynamic light scattering (DLS), where analysis was performed at 25 °C with an angle of detection of 180° with the heterodyne-backscatter arrangement. A total of 1 mL of nanosuspension was placed in a sample cell and the FLEX software was used to analyze electrophoretic mobility (for ZP) and particle size distribution using Brownian motion. For each nanosuspension sample, the mean value was recorded as an average of three measurements.

#### 2.2.3. Encapsulation Efficiency

The encapsulated drug was quantified by high-performance liquid chromatography (HPLC) using a C18 column, mobile phase of pH 3 phosphoric acid buffer: acetonitrile (50:50 (*v*/*v*)), a 1.2 mL/min isocratic flow rate, an injection volume of 20 µL and a 240 nm detection wavelength. TA was quantified using a calibration curve with demonstrated linearity in the range of 0.1–0.9 µg/mL. Based on the standard deviation (Sy) and slope (S) of the calibration curve, the limit of detection (LOD) and limit of quantification (LOQ) of TA was determined using the Equations (1) and (2).
(1)$$\mathrm{LOD}=3.3\left(\frac{Sy}{S}\right)$$
(2)$$\mathrm{LOQ}=10\;\left(\frac{Sy}{S}\right)$$

A fixed volume of drug-loaded NPs suspension was diluted with mobile phase and the encapsulation efficiency was calculated using the following Equation (3):(3)$$\mathrm{Encapsulation}\;\mathrm{efficiency}\;\left(\%\right)=\left(\frac{\mathrm{Drug}\;\mathrm{present}\;\mathrm{in}\;\mathrm{nanoparticles}}{\mathrm{total}\;\mathrm{drug}\;\mathrm{added}}\right)\;\times \;100$$

#### 2.2.4. Freeze-Drying of Nanoparticles

The NPs in suspension form containing 1% trehalose as cryoprotectant was pre-cooled at −20 °C overnight. Upon pre-cooling, the samples were placed in the freeze dryer (Freezone 2.5, LABCONCO, Kansas city, MO, USA) at 6 Pa for 72 h maintained at −50 °C. These freeze-dried NPs were used for thermal studies and FTIR analysis.

#### 2.2.5. Thermal Analysis

The thermal behavior of the formulation materials, PLGA and chitosan-coated PLGA NPs was investigated using thermal gravimetric analysis (TGA) and differential scanning calorimetry (DSC). Thermal analysis was performed using a nitrogen gas flow rate of 50 mL/min and a heating rate of 10 °C/min. Thermal decomposition analysis was performed from room temperature to 500 °C with a sample weight between 5 and 10 mg. The weight loss and onset of degradation were simultaneously recorded as a function of temperature/time. Resulting TGA thermograms were analyzed using TA universal analysis software. Calibration using indium was performed to ensure the accuracy and precision of the DSC thermograms obtained. Accurately weighed samples of 5–10 mg were loaded in T zero aluminum pans with a pinhole lid. The analysis was performed up to 350 °C under a nitrogen atmosphere at a heating rate of 10 °C/min.

#### 2.2.6. Fourier-Transform Infrared Spectroscopy Analysis

FTIR was performed on a Varian 600 IR using potassium bromide (KBr) disc with the spectra obtained in the region of 4000–600 cm^−1^ at a resolution of 2.0 cm^−1^ and 64 scans per run. The raw materials and freeze-dried nanoparticles were mixed with KBr in 1:10 ratios and compressed into a disc for analysis.

#### 2.2.7. In Vitro Drug Release

The drug release study was carried out using phosphate buffered saline (PBS (pH 7.4)) with 1% Tween^®^ 80 and 0.01% sodium azide as release medium. Before performing the drug release study, the solubility of TA in the release medium was determined, where 2 and 10 mg of TA was incubated in 5 mL of PBS and PBS with 1% Tween^®^ 80 at 37 °C with agitation. After 24 h the samples were centrifuged at 10,000 rpm and the supernatant was syringe filtered prior to HPLC analysis. Initially, release media containing nanosuspension with a drug concentration of 25 µg/mL was taken. The release medium was distributed into 2 mL microcentrifuge tubes containing nanosuspension equivalent to 50 µg of TA. These microcentrifuge tubes were placed in a shaker with continuous agitation at 37 °C. At every sampling point, a new microcentrifuge tube was taken and centrifuged at 15,000 rpm for 15 min and the drug present in the supernatant was quantified. The %drug release and %cumulative drug release were calculated by using the following Equations (4) and (5).
(4)$$\mathrm{Drug}\;\mathrm{release}\;\left(\%\right)=\left(\frac{\mathrm{Released}\;\mathrm{drug}}{\mathrm{total}\;\mathrm{drug}}\right)\;\times \;100$$
(5)$$\;\mathrm{Cumulative}\;\mathrm{drug}\;\mathrm{release}\;\left(\%\right)=\left(\frac{\mathrm{Volume}\;\mathrm{of}\;\mathrm{sample}\;\mathrm{withdrawn}}{\mathrm{bath}\;\mathrm{Volume}}\right)\;\times \;P(t-1)$$

P = %release at time ‘t’.

P (t − 1) = %release previous to ‘t’.

The best curve fit of the in vitro TA release was analyzed with the mathematical models: zero and first order, Hixson–Crowell, Higuchi and Korsmeyer–Peppas based on regression coefficient (*R*^2^) values.

#### 2.2.8. Statistical Analysis

All the data were presented as mean ± SD. Minitab statistical software was used to analyze the design of experiments responses.

## 3. Results

### 3.1. Screening of Chitosan-Coated PLGA Nanoparticles

This study is focused on developing a nanoparticulate system encapsulating a corticosteroid for topical ocular drug delivery to treat acquired retinal vasculopathies. PLGA NPs are potential candidates for drug delivery to the back of the eye diseases following topical instillation [25]. The surface modification of these NPs with mucoadhesive polymers would enhance the ocular residence time and hence increasing the drug’s bioavailability [44]. PLGA undergoes biodegradation by hydrolysis resulting in glycolic and lactic acids, which further enter the tricarboxylic acid cycle and is metabolized into carbon dioxide, energy, and water; this process makes PLGA in vivo degradable [45]. Biodegradation of chitosan occurs through hydrolysis by proteases (primarily lysozyme). The degradation products are monosaccharides or oligosaccharides and glycosaminoglycan or glycol-amino proteins, which are natural metabolites [46]. The goal of the current study is to optimize the formulation variables to attain the NPs with the smallest size possible, which can encapsulate the appropriate amounts of drug to suit the ocular application. Chitosan-coated PLGA NPs were prepared using a single emulsion technique by modifying two previously published methods [36,47]. Initially, screening experiments were performed to study the reproducibility of the chosen method and to test the effectiveness of PVA and PF-127 as stabilizers. The findings of the blank PLGA (A1 and A2) and chitosan-coated PLGA NPs (CS-A1 and CS-A2) prepared with 0.25% (*w*/*v*) PVA or 1% (*w*/*v*) PF-127 are represented in Table 2. PVA showed promising results as the stabilizer in these studies and was therefore considered for the screening experiments [36]. In our previous (unpublished) studies, PF-127 was used as the stabilizer, which produced stable NPs. It has also shown promising results in the past [48].

The NPs fabricated with PF-127 as stabilizer yielded smaller NPs of 200 ± 61.16 nm and 187 ± 23.55 nm with low PDI (0.06–0.07) and high encapsulation of the TA (34–59%) compared to NPs prepared with PVA (27–50%). This was perhaps due to the entanglement of PEO moieties of PF-127 with polymeric chains of PLGA leading to more compact NPs with high drug protection [49,50,51]. Salama et al. formulated fluocinolone acetonide-loaded NPs using PLGA and PF-127; they attained a particle size of 203 ± 5 nm with 56 ± 4% EE [23]. Along with the smaller size, the NPs also released the drug in a controlled manner up to 24 h; the authors postulated this was due to the interactions between PLGA and PEO groups of PF-127, which increases the stability of NPs (the entanglement of PLGA chains and PEO moieties could make the particle more compact and protect the drug leading to extended drug release). The encapsulation efficiency of the NPs decreased upon chitosan coating due to the repeated centrifugation step after chitosan coating, which may lead to loss of some particles and surface bound drug.

### 3.2. Optimization of Nanoparticles Using Statistical Experimental Design

DOE is widely used in the pharmaceutical industry and academia to investigate the factors that control the drug delivery system. An experimental design is important to study several factors at multiple levels using predefined experiments which can be applied to design a product or process [52]. DOE has proved to be a powerful tool for designing nanoparticulate systems, which can identify the effect of individual factors and interaction between factors on the characteristics of the NPs [53]. Box-Behnken design is more efficient and cost-effective than a central composite design with the same parameters due to fewer design points [54]. The central composite design usually consists of design points outside the region of interest (axial points outside the cube), in contrast the Box-Behnken design points lie in the selected region of interest (region of interest is selected based on screening experiments). The Box-Behnken, response surface statistical design consisting of three factors with three levels prioritized for this study to optimize the NPs prepared by the emulsion method. The model evaluates the main effects, interaction effects and quadratic effects of the variable factors on the NPs characteristics. It also identifies the combination of factors to obtain the predicted response based on the point prediction (response optimizer). Trabado et al. used a Box-Behnken response surface DOE to optimize the characteristics (particle size, ZP, PDI, %EE and %drug loading) of cyclosporin-loaded sorbitan ester NPs for topical ocular drug delivery [55]. In their study, the optimized formulation of 170.5 nm with a ZP of +33.9 mV and a drug loading capacity of 19.66% was achieved by using the response optimizer feature of the Box-Behnken DOE (by using the prediction composition generated by DOE they got the required responses for the chosen application). As mentioned in Section 2.2.1, in the present study, the three independent variables used were PLGA (3.5–4.5 mg/mL), PF-127 (1–2% (*w*/*v*)) and chitosan (1–2% (*w*/*v*)); the responses taken were particle size, polydispersity index (PDI), ZP and %EE. The lack of fit value of all the four responses generated by the DOE model used was *p* > 0.05 (insignificant), indicating the fit of experimental values to the model. The concentration of PLGA, chitosan and both combined have a significant effect on particle size (Figure 1). The low concentration of PLGA (3.5 mg/mL) and PF-127 (1% (*w*/*v*)) and medium concentration of chitosan (1.5% (*w*/*v*)) gave the smallest NPs of 334.95 ± 67.95 nm (Appendix A).

In a previous study, Madani et al. also observed an increase in nanoparticle size with the increase in PLGA concentration [56]. The authors postulated that in an emulsion technique, increased PLGA concentration in organic solvent elevates the viscosity of dispersed medium resulting in reduced shear stress leading to the formation of larger nanodroplets. There was no statistically significant effect of all three factors on the PDI of the NPs (Figure 2).

The PDI values of all the formulations ranged from 0.04 and 0.47 indicating that NPs prepared from an emulsion method were monodispersed (Figure 3). These results agree with recent study by Canioni et al., who formulated dexamethasone palmitate NPs to treat AMD via intravitreal injections [57]. They studied the influence of PEG-40-sterate, Pluronic^®^ F-168 and F-127 on NPs stability and monodispersion, the PF-127 at 1.5–2% (*w*/*v*) had shown best PDI values (0.02–0.12) with good stability compared to the other two surfactants (PDI: 0.03–0.5). This could be due to the arrangement of PF-127 on the nanoparticle surface with the PEO group facing the aqueous solvent creating steric stabilization. Thus, providing a protective barrier around the particle and preventing aggregation. The suitability of nanoformulations for specific drug delivery applications largely depends on average particle size and their dispersity. PDI is an important variable for studying the distribution of NPs or estimating their permeation efficiency across ocular barriers [58].

The particle size of all the PLGA NPs increased upon chitosan coating (Appendix A), this is potentially due to adsorption of the chitosan on the porous surface of the PLGA NPs. Guo et al. studied the adsorption mechanism of chitosan onto PLGA NPs prepared by an oil-in-water emulsion technique [42]. In their study, they noticed the increase in particle size of PLGA NPs from 261.5 to 972.7 nm with an increase in chitosan concentration (0.12–2.4 g/L). Based on adsorption isotherms, the authors proposed that the adsorption of chitosan onto PLGA NPs depends on the chitosan cationic nature, high surface energy and non-uniform microporous surface of PLGA NPs. The amount of TA present in NPs was determined using HPLC and TA showed linearity between 0.1 and 0.9 µg/mL with regression greater than 0.999. Based on the calibration curve, LOD and LOQ of TA were calculated using Equations (1) and (2) and found to be 0.026 and 0.079 µg/mL, respectively. In the present work, PLGA NPs with the emulsion codes E7 and E10 showed high encapsulation of drug, 66.04 ± 2.60% and 63.38 ± 2.30%, respectively. Upon chitosan coating, the same formulations displayed higher encapsulation of the drug compared to other formulations (54.89 ± 2.11 and 55.18 ± 3.70%, respectively). Using the surface plots shown in Figure 4, the effect of PLGA, PF-127 and chitosan on the %EE of the NPs was represented.

The formulations prepared with a low concentration of PF-127 (1% (*w*/*v*)) showed high encapsulation efficiency (66.04 ± 2.60%) whereas for chitosan-coated NPs, medium concentrations of chitosan (1.5% *w*/*v*) displayed a high %EE of 55.18 ± 3.70% (Figure 4). PLGA NPs, E7 and E10, showed smaller particle size (227.55 ± 18.60 and 274.80 ± 0.99 nm) with low PDI values (0.12 and 0.11). The particle size of CS-E7 and CS-E10 was 439 ± 35.36 and 334 ± 67.95 nm with PDI values of 0.23 and 0.15 and ZP of +24.15 ± 2.05 mV and +26.65 ± 9.97 mV, respectively. Tahara et al. fabricated PLGA NPs encapsulating coumarin-6 dye using emulsion solvent diffusion and also surface coated them with chitosan, glycol chitosan and polysorbate [25]. Their particle size of PLGA NPs was 200 nm and the surface-modified NPs ranged between 250 and 500 nm. When these formulations were applied topically on the mouse eye, after 30 min the entire eyeball (anterior segment, the cornea, iris/ciliary and the retina) was dyed, which was observed under the fluorescent microscope. According to their fluorescent images, the topical formulation may enter the back of the eye via trans-corneal, trans-scleral and uveal routes. This study highlights that topically applied nanoparticulate systems utilize multiple routes (trans-corneal, trans-scleral and uveal routes) to travel to the back of the eye. According to previously published research, the nanoparticulate systems having similar characteristics obtained in the present study permeated to the back of the eye when instilled topically [59,60]. 

### 3.3. Response Optimization Using Response Surface Design

Based on the responses of the DOE, the response optimizer predicts the combination of variables to get the desired responses of particle size, ZP and %EE for the NPs. Two predictions were entered into the software, and they were coded as PE1, PE2 and CS-PE1 and CS-PE2 for PLGA NPs and chitosan-coated PLGA NPs, respectively. For prediction one (PE1 and CS-PE1), the goal of achieving a smaller particle size of chitosan-coated NPs with high encapsulation of TA was entered (Appendix A). Whereas for prediction two (PE2 and CS-PE2) NPs with smaller particle size and ZP of +25 mV was considered as desired responses (Appendix A). The DOE model generated the composite desirability (CD) graphs for each factor and response to show the reason for selecting the specific concentrations of the factors to achieve the desired result (Appendix A). The CD value was 0.84 and 1 for predictions 1 and 2, if the CD value is close to one there are more chances of getting the prediction result. The NPs were prepared with the same procedure summarized in Section 2.2.1 and the characteristics obtained are represented in Table 3. Similar to the previous formulations prepared by the Box-Behnken DOE, the particle size increased upon chitosan-coating of the PLGA NPs. There was no significant difference in the particle size of the optimized formulations from DOE (CSE7 and CSE10) and the prediction emulsions (PE1 and PE2). However, the chitosan-coated prediction emulsions have lower PDI values (0.09–0.13) compared to CSE7 and CSE10 (0.15–0.23).

The experimental response values of the prepared prediction nano-emulsions were similar to the predicted response values of the software. The predicted particles size of the chitosan-coated PE1 and PE2 was 388.5 ± 46.7 nm, and the obtained particle size was 386.67 ± 15.14 and 351.33 ± 27.02, respectively (Appendix A). The obtained response values for both the predictions are within the standard error of the fit generated by the response optimizer suggesting the validity and fit of the model. The two prediction formulations and E10 formulation, before and after chitosan coating were selected for further studies based on their particle size, PDI, %EE and ZP. The great challenge with topical ocular drug delivery is to increase the drug’s bioavailability. The TA-loaded NPs fabricated in the current study might show enhanced bioavailability with an increase in the ocular residence compared to the free TA. Recently, Xing et al. prepared TA-loaded PLGA-chitosan NPs using an emulsion technique to treat ocular inflammation [61]. They investigated the pharmacokinetic profile of the free TA and TA-loaded NPs on the albino rabbits upon administration into the conjunctival sac. The maximum concentration of TA observed for free TA and TA-loaded NPs was 15.8 ± 0.57 and 43.2 ± 0.57 µg/L at 1 and 6 h, respectively in the aqueous humor. The free TA cleared after 6 h, whereas the TA-loaded NPs maintained the concentration until the end of the study (24 h). This study highlights the effectiveness of nanoformulations in increasing the bioavailability of TA.

### 3.4. Thermal Analysis

Thermal analysis was also performed for the NPs fabricated using an emulsion technique. This was carried out in order to study the stability of the materials in the formulations and to investigate the impact of nanoparticle formation on their stability (Figure 5). Extended thermal stability was noticed in the PLGA NPs (E10, PE1 and PE2) compared to drug, polymer, and surfactant (Appendix A).

In the thermal analysis of chitosan-coated NPs (CSE10, PE1 and PE2), the thermograms of these NPs (Appendix A) exhibited weight loss around 300 °C due to the loss of water at the initial degradation of the surface adsorbed chitosan [62]. Increased thermal stability of the chitosan-coated formulations was observed compared to polymer and drug alone, which was also noticed in the thermograms of PLGA NPs. In previous studies, similar thermal behavior was noticed where the NPs demonstrated enhanced thermal stability compared to the individual components probably due to their intact structure and molecular dispersion [63,64].

To examine the physical state of the drug and the molecular dispersion in the NPs, DSC analysis was performed (Figure 6). PLGA displayed two endothermic peaks at 47.58 and 299.91 °C corresponding to glass transition/relaxation and degradation, respectively [65]. PF-127 displayed an endothermic peak at 56.96 °C associated with the melting temperature [66]. The endothermic glass transition/relaxation peak of PLGA and melting peak of PF-127 were combined and shifted to 55.54 °C for E10 formulations, which suggests the interaction of the polymer and surfactant in the NPs. TA exhibited a sharp endothermic peak at 291.10 °C due to the melting point of the crystalline state [67]. Both the endothermic peaks of the drug (melting peak) and PLGA (degradation peak) displayed at 312.13 °C in the physical mixture, the intensity of the drug peak decreased in the physical mixture may be due to the dilution effect of polymer and surfactant (a similar amount of TA and PLGA was present in the formulation and the physical mixture). However, the degradation peak of PLGA was not seen in any of the formulations due to the enhanced thermal stability which was observed in the TGA results (Appendix A).

The sharp endothermic peak of the drug was not seen in any of the PLGA NPs (Appendix A), which indicates the change of the drug to the amorphous form leading to enhanced aqueous solubility of the drug and molecular dispersion in the nanoparticulate system [67]. These findings agree with Salama et al. where fluocinolone acetonide-loaded nanoparticles were prepared using PLGA and PF-127 [23]. They noticed the conversion of corticosteroid to amorphous form by loss of characteristic endothermic peak of the drug in formulations (which is present in pure drug thermogram due to its crystalline state). The fusion of endothermic peaks of PLGA and PF-127 highlights the entanglement of PLGA chains and PF-127 moieties indicating the formation of intact NPs as mentioned in Section 3.1.

### 3.5. Fourier Transform Infra-Red Spectroscopy (FT-IR) Analysis

FT-IR spectra and functional groups of PF-127, PLGA, TA, physical mixture and PLGA NPs (E10) are presented in Figure 7 and Table 4. Drug and excipients functional groups were present in the formulation with peaks at similar wavenumbers, which suggests that there was no interaction between the drug and the components [59].

Pure TA spectra displayed an infrared absorption band at 3398 cm^−1^ related to the hydrogen-bonded hydroxyl stretching vibration, at 1706 cm^−1^ associated with stretching vibration of the carbonyl group (in the aliphatic ester bonds). Another typical absorption band of TA at 1057 cm^−1^ is due to the stretching vibration of C–F. The physical mixture also showed the characteristic peaks of the TA in a similar wavelength range. Blueshift of the ketonic carbonyl and C–F groups was observed in the formulation at 1762 and 1060 cm^−1^, respectively.

The FT-IR spectra (Figure 8) suggested the existence of chitosan coating on PLGA NPs by the presence of characteristics functional group of chitosan; C–H, CONH_2_ and C–O on CS-E10 spectra at 2935, 1631 and 1087 cm^−1^, respectively, with a slight shift in wavenumber [62,71,72,73].

The FTIR characteristic peaks of chitosan together with the conversion of negative ZP to positive ZP and the thermogram changes of coated NPs suggest the successful coating of chitosan onto PLGA NPs. The characteristic negative charge of the eye structures can be utilized to enhance drug bioavailability by introducing oppositely charged polymer for surface coating [74].

### 3.6. In Vitro Drug Release Study

The %cumulative drug release was plotted against time (h) for the in vitro drug release study performed using PBS (pH 7.4) with 0.01% sodium azide and 1% Tween^®^ 80 as release media. To increase the solubility of hydrophobic TA, Tween^®^ 80 was used as a surfactant in the release medium [75]. The surfactant increased the solubility of the drug from 21 ± 2.16 to 102.05 ± 4.78 µg/mL. To avoid bacterial contamination of the formulation and the release media over 70 h, 0.01% sodium azide was used as an antibacterial agent [36].

The drug release from the PLGA nano-formulations reached a plateau after 23 h and the chitosan-coated nano-formulations reached a plateau at 27 h. Even though the release from all the three PLGA NPs was similar, the CS-E10 formulation showed more drug release compared to other formulations (59.47 ± 3.28%). This could be due to a lower concentration of PLGA in E10 (3.5 mg/mL) compared to PE1 (3.7 mg/mL) and PE2 (4.5 mg/mL). With the increase in polymer concentration, the drug might have better protection leading to a delay in the drug release [76]. The chitosan-coated NPs fabricated using an emulsion method controlled the initial burst release when compared to PLGA NPs. In the first 2 h, PLGA NPs released 44–40% of the drug (potentially due to the release of the drug close to the pores of the PLGA NPs) whereas chitosan-coated PLGA NPs released 21–37% of the drug, which indicates the role of chitosan coating in controlling the initial burst release (Figure 9). Similar control of initial burst release was observed in previously published findings of NPs prepared with the same polymers [62,77].

The NPs demonstrated a biphasic release, burst release on Day 1 followed by controlled release; this release pattern is in agreement with previous investigations [78,79,80]. Pandit et al. formulated bevacizumab-loaded, chitosan-coated PLGA NPs using emulsion technique and investigated the in vitro drug release profile in PBS [36]. They observed a similar burst release of the drug in the first day followed by controlled release over 72 h reaching a plateau. The curve fit of the in vitro TA release was analyzed with the mathematical models considering regression coefficient (*R*^2^) as represented in Table 5. Korsmeyer–Peppas is most fitting to the TA release from the NPs (*R*^2^ between 0.621 and 0.894) when compared to other models, which suggests the sustained release of the TA. This model indicates that the release from the PLGA NPs involves multiple release mechanisms, such as diffusion, dissolution and swelling [81].

Commercially available corticosteroid implants for posterior inflammation like Ozurdex^®^ (dexamethasone intravitreal implant) and Retisert^®^ (fluocinolone acetonide intravitreal implant) release about 0.2 µg of drug per day [82]. The anti-VEGF activity of TA was observed on human retinal epithelial cell lines from a concentration of 4.34 µg [9]. In the present study, around 25 µg of TA was released in the first two days leading to a sustained release. The released drug is more than the therapeutic dose of the similar corticosteroid intravitreal implants, which is advantageous considering the loss of the drug during the transport from the front of the eye to the posterior segment. The drug dose regimen depends on the pathological condition of the disease. All three optimized nano-emulsions (E10, PE1 and PE2) before and after chitosan coating exhibited controlled release of the drug over the duration of the experiment (70 h) due to the release mechanism of PLGA NPs.

## 4. Conclusions

The formulation of chitosan-coated PLGA NPs was successfully optimized with a Box-Behnken response surface DOE and a response optimizer. The optimized surface-modified NPs (CS-E10, CS-PE1 and CS-PE-2) were reproducible and colloidally stable with a particle size of 334 ± 67.95 to 386 ± 15.14 nm and PDI between 0.09 ± 0.04 and 0.15 ± 0.08, having a ZP between +26 ± 9.97 and +33 ± 4.69 mV. These NPs encapsulated 55–57% of TA and displayed a controlled release of the drug reaching a plateau in 27 h. This study highlighted the importance of PF-127 as a surfactant in yielding smaller particles with uniform distribution and the utilization of DOE and response optimizer in achieving the desired NPs. The prediction formulations (CS-PE1 and CS-PE2) obtained from the response optimizer showed high %EE and controlled release of the drug with monodispersed NPs. The polymeric matrix of PLGA supports controlled diffusion of encapsulated drug, while the mucoadhesive property of chitosan may enhance permeation across the barriers of the eye. The size of the NPs in conjunction with the biodegradable and biocompatible properties of the polymers suggest these particles might be promising for topical ocular drug delivery, which could improve patient comfort and outcomes while reducing healthcare expenses by negating or reducing the need for intravitreal injections.

## Figures and Tables

**Figure 1 pharmaceutics-13-01590-f001:**
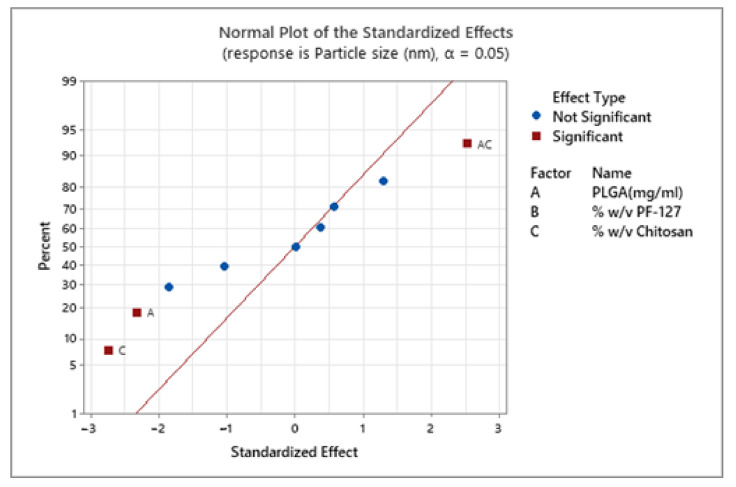
Effect of factors (PLGA, PF-127 and chitosan) on the particle size of chitosan-coated PLGA nanoparticles. The blue points represent the insignificant factors, and the red points represent significant factors.

**Figure 2 pharmaceutics-13-01590-f002:**
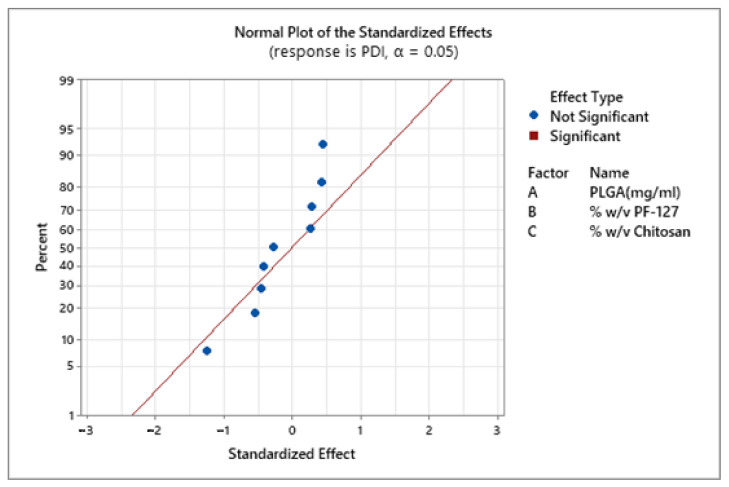
Effect of PLGA, PF-127 and chitosan on PDI of the prepared chitosan-coated PLGA nanoparticles. The blue points represent the insignificant factors.

**Figure 3 pharmaceutics-13-01590-f003:**
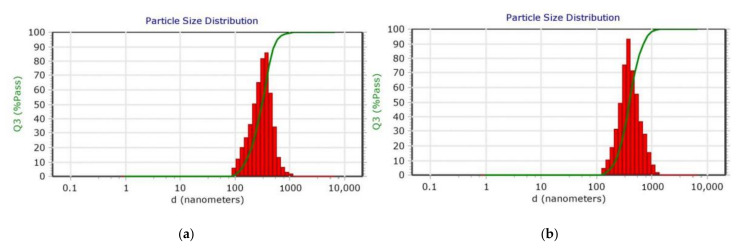
Dynamic light scattering (DLS) particle size distribution of (**a**) PLGA nanoparticles (E10) and (**b**) chitosan-coated PLGA nanoparticles (CS-E10). The graph represents the volume distribution of the nanoparticles, %Volume passing (Q3%pass) vs. diameter (*d*).

**Figure 4 pharmaceutics-13-01590-f004:**
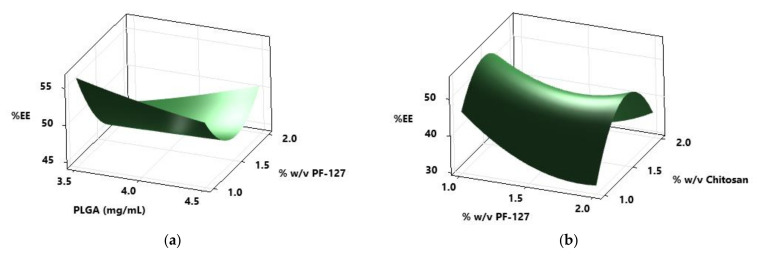
Response surface plots of (**a**) %EE (encapsulation efficiency) vs. PLGA and PF-127 (**b**) % EE vs. % (*w*/*v*) chitosan and % (*w*/*v*) PF-127.

**Figure 5 pharmaceutics-13-01590-f005:**
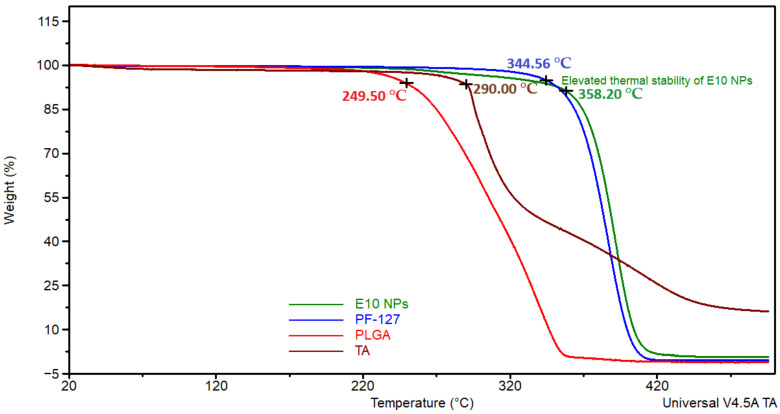
Thermogravimetric analysis (TGA) curves exhibiting the extended thermal stability of PLGA NPs (E10) compared to individual components (TA, PLGA and PF-127).

**Figure 6 pharmaceutics-13-01590-f006:**
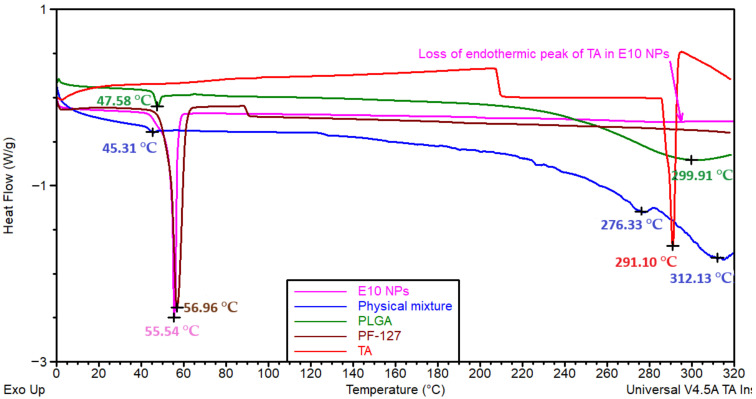
Differential scanning calorimetry (DSC) thermograms of physical mixture, individual components (TA, PLGA and PF-127) and PLGA NPs (E10).

**Figure 7 pharmaceutics-13-01590-f007:**
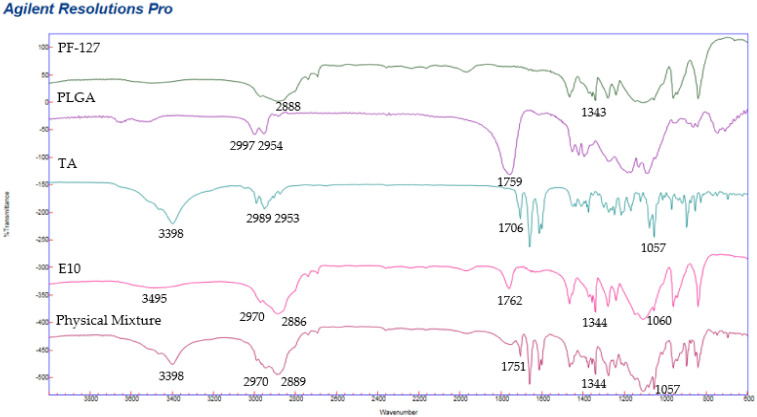
FTIR spectrums of individual components (TA, PLGA and PF-127), physical mixture and PLGA nanoparticles (E10).

**Figure 8 pharmaceutics-13-01590-f008:**
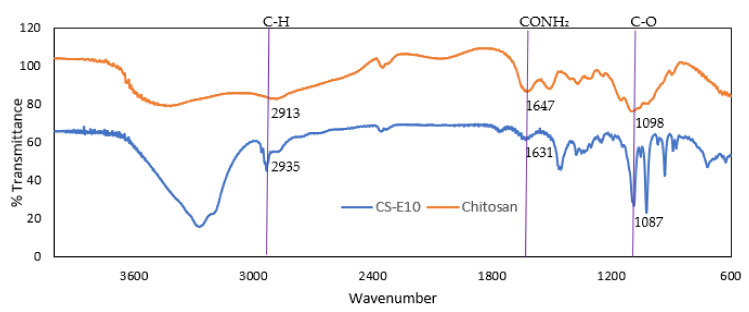
FTIR spectra of chitosan and chitosan-coated PLGA NPs (CS-E10) highlighting the functional groups.

**Figure 9 pharmaceutics-13-01590-f009:**
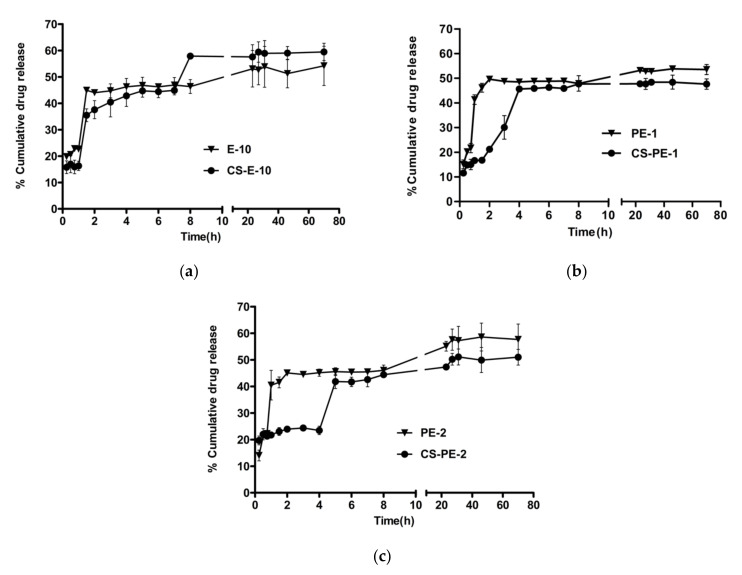
Comparison of in vitro drug release of non-coated and coated NPs: (**a**) E-10 and CS-E10, (**b**) PE-1 and CS-PE-1 and (**c**) PE-2 and CS-PE-2. The concentration of TA-loaded NPs in release media was 25 µg/mL for all formulations. Data = mean ± std (*n* = 3).

**Table 1 pharmaceutics-13-01590-t001:** Compositions of the chitosan-coated PLGA NPs with formulation codes for the experiments generated by DOE.

Emulsion Code	PLGA (mg/mL)	PF-127% (*w*/*v*)	Chitosan% (*w*/*v*)
CS-E1	4	2	2
CS-E2	4.5	1.5	2
CS-E3	3.5	1.5	2
CS-E4	3.5	2	1.5
CS-E5	4	1	1
CS-E6	3.5	1.5	1
CS-E7	4.5	1	1.5
CS-E8	4.5	2	1.5
CS-E9	4	1.5	1.5
CS-E10	3.5	1	1.5
CS-E11	4.5	1.5	1
CS-E12	4	2	1
CS-E13	4	1	2

**Table 2 pharmaceutics-13-01590-t002:** Particle size, zeta potential, PDI and % encapsulation efficiency of the NPs from screening experiments. Particle size is the representation of the diameter of the particle (nm). Data = mean ± std (*n* = 3).

Emulsion Code	Stabilizer (% (*w*/*v*))	Particle Size (nm)	Zeta Potential (mV)	PDI
A1	PVA (0.25)	359 ± 76.37	−14 ± 5.44	0.15 ± 0.07
CS-A1	PVA (0.25)	346 ± 125.37	+18 ± 1.77	0.70 ± 0.53
A2	PF-127 (1)	200 ± 61.16	−20 ± 9.26	0.07 ± 0.01
CS-A2	PF-127 (1)	187 ± 23.55	+14 ± 5.44	0.06 ± 0.01

**Table 3 pharmaceutics-13-01590-t003:** Characteristics of the PLGA and chitosan-coated PLGA nanoparticles prepared by point prediction using response surface DOE. Particle size is the representation of the diameter of the particle (nm). Data = mean ± std (*n* = 3).

Emulsion Code	Particle Size (nm)	PDI	Zeta Potential (mV)	%Encapsulation Efficiency
PE1	318.23 ± 18.61	0.220 ± 0.08	−7.4 ± 2.43	64.80 ± 3.95
CS-PE1	386.67 ± 15.14	0.136 ± 0.05	+33.3 ± 4.69	57.14 ± 3.81
PE2	240.47 ± 48.75	0.084 ± 0.05	−6.9 ± 3.08	60.31± 2.46
CS-PE2	351.33 ± 27.02	0.098 ± 0.04	+31.97 ± 0.21	51.71 ± 1.82

**Table 4 pharmaceutics-13-01590-t004:** Representation of FT-IR peaks and functional groups of the individual components, physical mixture and PLGA NPs (E10).

Frequency (cm^−1^)	Functional Group	Individual Components (cm^−1^)			Physical Mixture (cm^−1^)			E10-Nanoparticles (cm^−1^)			Ref.
		PLGA	PF-127	TA	PLGA	PF-127	TA	PLGA	PF-127	TA	
3510	OH	3520						3495			[68]
3398	OH			3398			3398				[69]
3000–2950	C–H	2997–2954		2989–2953			2970–2889	2970–2886			[68]
2891	C–H		2888			2889			2886		[70]
2985–2937	C–H			2989–2951			2970–2889			2970–2886	[69]
1705	C=O			1706						1762	[69]
1757	C=O	1759			1751			1762			[68]
1344	O–H		1343			1344			1344		[70]
1055	C–F			1057			1057			1060	[67]

**Table 5 pharmaceutics-13-01590-t005:** In vitro drug release mathematical model fitting concerning linear regression coefficient (*R*^2^).

Formulation Code	Regression Coefficient (*R*^2^)				
	Zero-Order	First-Order	Hixson–Crowell	Higuchi	Korsmeyer–Peppas
PE 1	0.552	0.582	0.572	0.69	0.791
PE 2	0.756	0.767	0.763	0.893	0.894
E 10	0.338	0.41	0.386	0.464	0.621
CS-PE1	0.237	0.252	0.247	0.387	0.641
CS-PE2	0.503	0.541	0.528	0.676	0.845
CS-E10	0.504	0.586	0.56	0.681	0.853

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
