# Peer review of "Chitosan-Coated PLGA Nanoparticles Encapsulating Triamcinolone Acetonide as a Potential Candidate for Sustained Ocular Drug Delivery"

_pharmaceutics, 2021, doi:10.3390/pharmaceutics13101590_

Round 1
Reviewer 1 Report
The current manuscript is entitled "Topical ocular drug delivery through surface-modified nanoparticles to treat the posterior segment eye diseases". On the one hand, according to the authors in the abstract, "this research aims to develop a topically applied nanoparticulate system encapsulating a corticosteroid for extended drug release". However, on the other hand, in the introduction, the authors say that "the current research was focused on developing a topically applied nanoparticulate system exhibiting extended drug release for the treatment of posterior segment disease." The study did not use research methods that unequivocally show the possibility of the proposed drug delivery system to be used in the treatment of diseases of the posterior segment of the eye. In this regard, the authors must change the article's title to reflect the actual results achieved.
PVA and DCM used should be added in section 2.1. Materials.
FTIR analysis has been performed with freeze-dried NPs. Therefore, the authors should include the freeze-drying process in 2.2. Methods. Also, it is necessary to say in which studies lyophilized nanoparticles were used. Furthermore, the authors should explain the statistical analysis in the same section.
The authors claim that the proposed nanocarriers are stable. But there is no mention under what conditions and for how long they are stable. Authors should be more specific and not draw conclusions based on research by other authors.
I would advise the authors to discuss triamcinolone acetonide-related studies, such as https://www.tandfonline.com/doi/full/10.1080/21691401.2021.1895184.
Author Response
- The current manuscript is entitled "Topical ocular drug delivery through surface-modified nanoparticles to treat the posterior segment eye diseases". On the one hand, according to the authors in the abstract, "this research aims to develop a topically applied nanoparticulate system encapsulating a corticosteroid for extended drug release". However, on the other hand, in the introduction, the authors say that "the current research was focused on developing a topically applied nanoparticulate system exhibiting extended drug release for the treatment of posterior segment disease." The study did not use research methods that unequivocally show the possibility of the proposed drug delivery system to be used in the treatment of diseases of the posterior segment of the eye. In this regard, the authors must change the article's title to reflect the actual results achieved.
Author Response: As suggested the title has been updated to “Chitosan-coated PLGA nanoparticles encapsulating triamcinolone acetonide as a potential candidate for sustained ocular drug delivery”. Chitosan-coated nanoparticles were developed and optimized in the present study. The ultimate goal is to investigate the feasibility of this system for topical ocular drug delivery with extended drug release. As such, the title stated the chitosan-coated PLGA nanoparticles as a potential candidate for sustained ocular drug delivery.
- PVA and DCM used should be added in section 2.1. Materials
Author response: Updated the 2.1 material section and included PVA and DCM in lines 120-121.
- FTIR analysis has been performed with freeze-dried NPs. Therefore, the authors should include the freeze-drying process in 2.2. Methods. Also, it is necessary to say in which studies lyophilized nanoparticles were used. Furthermore, the authors should explain the statistical analysis in the same section.
Author response: As suggested in Section 2.2.4 – Freeze-drying of nanoparticles was added and mentioned the studies where lyophilized nanoparticles were used in lines 176-180. Statistical analysis Section 2.2.8 was also added to the methods section in lines 219-221.
Updated text:
2.2.4 Freeze-drying of nanoparticles
The nanoparticles in suspension form containing 1% trehalose as cryoprotectant was pre-cooled at -20 ℃ overnight. Upon pre-cooling, the samples were placed in the freeze dryer (Freezone 2.5, LABCONCO, US) at 6 Pa for 72 h maintained at -50 ℃. These freeze-dried NPs were used for thermal studies and FTIR analysis.
2.2.8 Statistical analysis
All the data were presented as mean±SD. Minitab statistical software was used to analyse the design of experiments responses.
- The authors claim that the proposed nanocarriers are stable. But there is no mention under what conditions and for how long they are stable. Authors should be more specific and not draw conclusions based on research by other authors.
Author response: The optimized NPs in the current study have the zeta potential values between +26 mV and +33 mV; zeta potential values greater than +25 mV or less than -25 mV known to have high stability due to the repulsive forces, which can be used to predict the colloidal stability and agglomerative nature of the NPs [1]. The low zeta potential values lead to aggregation between the particles due to Van der Waals interparticle interactions.
Accordingly, the term stable was updated to “colloidally stable” in the conclusion (line number-538)
- I would advise the authors to discuss triamcinolone acetonide-related studies, such as https://www.tandfonline.com/doi/full/10.1080/21691401.2021.1895184.
Author response: Thank you for highlighting this recent study, which is relevant to the present work. As suggested, the study was discussed in terms of the bioavailability of the TA-loaded NPs in the eye (in Section 3.3 from line numbers 390 to 401).
Updated text:
The great challenge with topical ocular drug delivery is to increase the drug's bioavailability. The TA-loaded NPs fabricated in the current study might show enhanced bioavailability with an increase in the ocular residence compared to the free TA. Recently, Xing et al. prepared TA-loaded PLGA-chitosan NPs using an emulsion technique to treat ocular inflammation [2]. They investigated the pharmacokinetic profile of the free TA and TA-loaded NPs on the albino rabbits upon administration into the conjunctival sac. The maximum concentration of TA observed for free TA and TA-loaded NPs was 15.8 ± 0.57 µg/L and 43.2 ± 0.57 µg/L at 1 h and 6 h, respectively in the aqueous humour. The free TA cleared after 6 h, whereas the TA-loaded NPs maintained the concentration until the end of the study (24 h). This study highlights the effectiveness of nanoformulations in increasing the bioavailability of TA.
References:
[1] G. Zografi, S. R. Byrn, and X. Chen, Solid-State Properties of Pharmaceutical Materials. Wiley, 2017.
[2] Y. Xing, L. Zhu, K. Zhang, T. Li, and S. Huang, “Nanodelivery of triamcinolone acetonide with PLGA-chitosan nanoparticles for the treatment of ocular inflammation,” Artif. Cells, Nanomedicine Biotechnol., vol. 49, no. 1, pp. 308–316, 2021, doi: 10.1080/21691401.2021.1895184.
Reviewer 2 Report
This manuscript studies the PLGA nanoparticles with surfactant medication using chitosan and loaded with Triamcinolone for proposed ocular application. The design of experiments (DOE) was applied to optimize formulation with suitable characteristics. The drug loading, size characteristics, interactions, calorimetry (DSC), and drug release are shown.
In general, the research design is poor, language needs improvement, and the presentation of figures must be improved in both quality and style.
- The title of the paper and the introduction talks about ocular delivery. I believe there should be some experimental way to justify the suitability of these particles for ocular delivery.
As no data significantly supports ocular application, the term ‘Ocular’ should be removed from the title to avoid misleading.
- Please reduce the background in the abstract and make it more relevant to the need for PLGA formulation than other delivery systems available.
- Several other forms of delivery systems have shown better delivery and control for the same drug. I suggest discussing other such systems in this manuscript, for eg. Liposomes.
- Similar research must be discussed and cited. For e.g. https://www.sciencedirect.com/science/article/pii/S1818087617300387
- Abstract: Please mention size as average size (radius or diameter) with SD at the appropriate places
- What was the purpose of adding sodium azide in the release study? I understand that sodium azide is broad-spectrum anti-microbial, usually added with release medium when tissues or biological samples are involved.
Formulation pH is critical for biological applications. Please include the pH of all formulations.
- What was the pH of PBS used in the release study?
- Please discuss and correlate the release with the required therapeutic dose of a drug.
- Table 2: PVA is rarely referred as surfactant, but is primarily referred as a stabiliser. I suggest replacing the term surfactant with stabilizer.
- Table 2: the term emulsion can be confusing, as data is for the final formulation in nanoparticles form.
- Line 212, Please include SD of the particle size with the average value.
- The caption of all figures and tables should be improved. Currently, the information appears incomplete.
- Section 3.6: Adding release modeling data will give a better understanding and increase interest.
- What benefit is achieved with the use of DOE?
- Please discuss excretion and clearance of PLGA from the ocular system. How long will it take to clear? What is the effect of repeated instillation of PLGA nanoparticles into eyes such as accumulation in eyes? How will it be cleared or excreted from eyes?
likewise, for chitosan - Most of figure's presentations, clarity and resolution must be improved.
- Figure 8: please also include lines with bullets for better presentation.
Author Response
1. The title of the paper and the introduction talks about ocular delivery. I believe there should be some experimental way to justify the suitability of these particles for ocular delivery. As no data significantly supports ocular application, the term ‘Ocular’ should be removed from the title to avoid misleading
Author response: The title has been updated to “Chitosan-coated PLGA nanoparticles encapsulating triamcinolone acetonide as a potential candidate for sustained ocular drug delivery”. Chitosan-coated nanoparticles were developed and optimized in the present study. The ultimate goal is to investigate the feasibility of this system for topical ocular drug delivery with extended drug release. As such, the title stated the chitosan-coated PLGA nanoparticles as a potential candidate for sustained ocular drug delivery.
2. Please reduce the background in the abstract and make it more relevant to the need for PLGA formulation than other delivery systems available.
Author response: As suggested the background has been reduced and highlighted the importance of PLGA and chitosan (Lines: 12-19).
Updated text:
The current treatment for the acquired retinal vasculopathies involves lifelong repeated intravitreal injections of either anti-vascular endothelial growth factor (VEGF) therapy or modulation of inflammation with steroids. Consequently, any treatment modification that decreases this treatment burden for patients and doctors alike would be a welcome intervention. To that end, this research aims to develop a topically applied nanoparticulate system encapsulating a corticosteroid for extended drug release. Poly (lactic-co-glycolic acid) (PLGA) nanoparticles (NPs) supports the controlled release of the encapsulated drug, while surface modification of these NPs with chitosan might prolong the mucoadhesion ability leading to improved bioavailability of the drug.
3. Several other forms of delivery systems have shown better delivery and control for the same drug. I suggest discussing other such systems in this manuscript, for eg. Liposomes.
Author response: As recommended, other nanoparticulate systems were mentioned in Section 1 from lines 65-73.
Updated text:
Different forms of nanomaterials being tested for anterior and posterior segment eye diseases include: NPs [1], nano-micelles [2], liposomes [3], dendrimers [4], nanoparticle-loaded contact lenses [5] and sub-conjunctival implants with nanostructures [6], etc. Liposomes and micelles have limitations such as low stability, leakage of entrapped drug and premature release of the loaded drug [7,8]. The changes in NPs-loaded contact lenses during the storage period like burst release of the drug before administration, swelling and optical transparency, etc., still need to be addressed [9]. For elderly patients wearing contact lenses and inserting implants with minor surgery leads to poor patient compliance. Recently, polymeric NPs has been studied for topical ocular delivery and shown promising results [10–12].
4. Similar research must be discussed and cited. For e.g., https://www.sciencedirect.com/science/article/pii/S1818087617300387
Author response: Thank you for suggesting, this paper was cited and discussed in Section 3.2 from lines 347-358.
Updated text:
Tahara et al. fabricated PLGA NPs encapsulating coumarin-6 dye using emulsion solvent diffusion also surface coated them with chitosan, glycol chitosan and polysorbate [12]. Their particle size of PLGA NPs was 200 nm and the surface-modified NPs ranged between 250 nm to 500 nm. When these formulations were applied topically on the mouse eye, after 30 min the entire eyeball (anterior segment, the cornea, iris/ciliary and the retina) was dyed, which was observed under the fluorescent microscope. According to their fluorescent images, the topical formulation may enter the back of the eye via trans-corneal, trans-scleral and uveal routes. This study highlights that topically applied nanoparticulate systems utilise multiple routes (trans-corneal, trans-scleral and uveal routes) to travel to the back of the eye. According to previously published research, the nanoparticulate systems having similar characteristics obtained in the present study permeated to the back of the eye when instilled topically [13,14].
5. Abstract: Please mention size as average size (radius or diameter) with SD at the appropriate places
Author response: Updated and added the term diameter with SD in the abstract (line number 22).
6. What was the purpose of adding sodium azide in the release study? I understand that sodium azide is broad-spectrum anti-microbial, usually added with release medium when tissues or biological samples are involved.
Author response: 0.01 % sodium azide was added to prevent the bacterial contamination of release media containing PBS and 1% Tween® 80.
7. Formulation pH is critical for biological applications. Please include the pH of all formulations.
What was the pH of PBS used in the release study?
Author response: All the formulations were redispersed in Milli-Q® Water (pH 7) and was updated in the line numbers 136-137. The pH of PBS used in the release study was 7.4 and has been added in line numbers 200 and 480.
8. Please discuss and correlate the release with the required therapeutic dose of a drug.
Author response: The discussion on the therapeutic dose of the drug has been added from lines 523-531.
Updates text:
Commercially available corticosteroid implants for posterior inflammation like Ozurdex® (dexamethasone intravitreal implant) and Retisert® (Fluocinolone acetonide intravitreal implant) release about 0.2 µg of drug per day [15]. The anti-VEGF activity of TA was observed on human retinal epithelial cell lines from a concentration of 4.34 µg [16]. In the present study, around 25 µg of TA was released in the first two days leading to a sustained release. The released drug is more than the therapeutic dose of the similar corticosteroid intravitreal implants, which is advantageous considering the loss of the drug during the transport from the front of the eye to the posterior segment. The drug dose regimen depends on the pathological condition of the disease.
9. Table 2: PVA is rarely referred as surfactant but is primarily referred as a stabiliser. I suggest replacing the term surfactant with stabilizer.
Author response: The term surfactant was replaced with a stabilizer as both have a similar function of dispersing the nanoparticles.
10. Table 2: the term emulsion can be confusing, as data is for the final formulation in nanoparticles form.
Author response: The term nanoemulsions was changed to nanoparticles in Table 2 caption.
11. Line 212, Please include SD of the particle size with the average value.
Author response: Included the SD values in lines 344 and 345.
12. The caption of all figures and tables should be improved. Currently, the information appears incomplete.
Author response: The captions of the figures and tables have been updated.
13. Section 3.6: Adding release modelling data will give a better understanding and increase interest.
Author response: As suggested, the Section 3.6 was updated with the mathematical release models (line number: 511-522).
Updated text: The curve fit of the in-vitro TA release was analysed with the mathematical models considering regression coefficient (R2) as represented in Table 5. Korsmeyer-Peppas is most fitting to the TA release from the NPs (R2 between 0.621 and 0.894) when compared to other models, which suggests the sustained release of the TA. This model indicates that the release from the PLGA NPs involves multiple release mechanisms such as, diffusion, dissolution and swelling [17].
Table 5: In-vitro drug release mathematical model fitting concerning linear regression coefficient (R2).
|
Formulation Code |
Regression coefficient (R²) |
||||
|
Zero-order |
First-order |
Hixson-Crowell |
Higuchi |
Korsmeyer-Peppas |
|
|
PE 1 |
0.552 |
0.582 |
0.572 |
0.69 |
0.791 |
|
PE 2 |
0.756 |
0.767 |
0.763 |
0.893 |
0.894 |
|
E 10 |
0.338 |
0.41 |
0.386 |
0.464 |
0.621 |
|
CS-PE1 |
0.237 |
0.252 |
0.247 |
0.387 |
0.641 |
|
CS-PE2 |
0.503 |
0.541 |
0.528 |
0.676 |
0.845 |
|
CS-E10 |
0.504 |
0.586 |
0.56 |
0.681 |
0.853 |
14. What benefit is achieved with the use of DOE?
Author response: The benefit of DOE and the chosen response surface model was discussed in Section 3.2 from line number 264-274.
Updated text:
DOE is widely used in the pharmaceutical industry and academia to investigate the factors that control the drug delivery system. An experimental design is important to study several factors at multiple levels using predefined experiments which can be applied to design a product or process [18]. DOE has proved to be a powerful tool for designing nanoparticulate systems, which can identify the effect of individual factors and interaction between factors on the characteristics of the NPs [19]. Box-Behnken design is more efficient and cost-effective than a central composite design with the same parameters due to less design points [20]. The central composite design usually consists of design points outside the region of interest (axial points outside the cube) in contrast the Box-Behnken design points lie in the selected region of interest (region of interest is selected based on screening experiments).
15. Please discuss excretion and clearance of PLGA from the ocular system. How long will it take to clear? What is the effect of repeated instillation of PLGA nanoparticles into eyes such as accumulation in eyes? How will it be cleared or excreted from eyes?
likewise, for chitosan
Author response: The clearance of PLGA/biodegradation was discussed in Section 3.2 from lines 229-234 and the pharmacokinetic profile was discussed in lines 400 to 406.
Updated text: PLGA undergoes biodegradation by hydrolysis resulting in glycolic and lactic acids, which further enter the tricarboxylic acid cycle and is metabolized into carbon dioxide, energy, and water; this process makes PLGA in-vivo degradable [21]. Biodegradation of chitosan occurs through hydrolysis by proteases (primarily lysozyme). The degradation products are monosaccharides or oligosaccharides and glycosaminoglycan or glycol-amino proteins, which are natural metabolites [22]
16. Most of figure's presentations, clarity and resolution must be improved.
Author response: As suggested the presentation and clarity of figures has been improved.
17. Figure 8: please also include lines with bullets for better presentation
Author response: The line graph for the in-vitro drug release was added in line number 486.
References:
[1] Z. Shi, S. K. Li, P. Charoenputtakun, C. Y. Liu, D. Jasinski, and P. Guo, “RNA nanoparticle distribution and clearance in the eye after subconjunctival injection with and without thermosensitive hydrogels,” J Control Release, vol. 270, pp. 14–22, 2018, doi: 10.1016/j.jconrel.2017.11.028.
[2] C. Li, R. Chen, M. Xu, J. Qiao, L. Yan, and X. D. Guo, “Hyaluronic acid modified MPEG-b-PAE block copolymer aqueous micelles for efficient ophthalmic drug delivery of hydrophobic genistein,” Drug Deliv, vol. 25, no. 1, pp. 1258–1265, 2018, doi: 10.1080/10717544.2018.1474972.
[3] G. Tan et al., “Bioadhesive chitosan-loaded liposomes: A more efficient and higher permeable ocular delivery platform for timolol maleate,” Int J Biol Macromol, vol. 94, no. Pt A, pp. 355–363, 2017, doi: 10.1016/j.ijbiomac.2016.10.035.
[4] M. Lancina, “Fast Dissolving Dendrimer Nanofiber (DNF) Mats as Alternative to Eye Drops for More Efficient Topical Antiglaucoma Drug Delivery,” ACS Biomater. Sci. Eng., vol. 3, no. 8, pp. 1861–1868, 2017.
[5] M. Y. Bin Sahadan et al., “Phomopsidione nanoparticles coated contact lenses reduce microbial keratitis causing pathogens,” Exp Eye Res, vol. 178, pp. 10–14, 2018, doi: 10.1016/j.exer.2018.09.011.
[6] K. McAvoy, D. Jones, and R. R. S. Thakur, “Synthesis and Characterisation of Photocrosslinked poly(ethylene glycol) diacrylate Implants for Sustained Ocular Drug Delivery,” Pharm Res, vol. 35, no. 2, p. 36, 2018, doi: 10.1007/s11095-017-2298-9.
[7] A. A. Alexander-bryant and W. S. Vanden Berg-foels, Bioengineering Strategies for Designing Targeted Cancer Therapies, 1st ed., vol. 118. Copyright © 2013 Elsevier Inc. All rights reserved., 2013.
[8] A. Akbarzadeh, R. Rezaei-sadabady, S. Davaran, S. W. Joo, and N. Zarghami, “Liposome : classification , preparation , and applications,” Nanoscale Res. Lett., vol. 8, no. 1, p. 1, 2013, doi: 10.1186/1556-276X-8-102.
[9] X. Zhang, X. Cao, and P. Qi, “Therapeutic contact lenses for ophthalmic drug delivery: major challenges,” J. Biomater. Sci. Polym. Ed., vol. 31, no. 4, pp. 549–560, 2020, doi: 10.1080/09205063.2020.1712175.
[10] A. H. Salama, A. A. Mahmoud, and R. Kamel, “A Novel Method for Preparing Surface-Modified Fluocinolone Acetonide Loaded PLGA Nanoparticles for Ocular Use: In Vitro and In Vivo Evaluations,” AAPS PharmSciTech, vol. 17, no. 5, pp. 1159–1172, 2016, doi: 10.1208/s12249-015-0448-0.
[11] N. Khan, Ameeduzzafar, K. Khanna, A. Bhatnagar, F. J. Ahmad, and A. Ali, “Chitosan coated PLGA nanoparticles amplify the ocular hypotensive effect of forskolin: Statistical design, characterization and in vivo studies,” Int. J. Biol. Macromol., 2018, doi: 10.1016/j.ijbiomac.2018.04.122.
[12] K. Tahara, K. Karasawa, R. Onodera, and H. Takeuchi, “Feasibility of drug delivery to the eye’s posterior segment by topical instillation of PLGA nanoparticles,” Asian J. Pharm. Sci., 2017, doi: 10.1016/j.ajps.2017.03.002.
[13] A. Tatke et al., “In situ gel of triamcinolone acetonide-loaded solid lipid nanoparticles for improved topical ocular delivery: Tear kinetics and ocular disposition studies,” Nanomaterials, 2019, doi: 10.3390/nano9010033.
[14] L. R. Schopf et al., “Topical ocular drug delivery to the back of the eye by mucus-penetrating particles,” Transl. Vis. Sci. Technol., 2015, doi: 10.1167/tvst.4.3.11.
[15] L. R. Steeples, N. P. Jones, and I. Leal, “Evaluating the Safety , Efficacy and Patient Acceptability of Intravitreal Fluocinolone Acetonide ( 0 . 2mcg / Day ) Implant in the Treatment of Non-Infectious Uveitis Affecting the Posterior Segment,” 2021.
[16] A. Hirani, A. Grover, Y. W. Lee, Y. Pathak, and V. Sutariya, “Triamcinolone acetonide nanoparticles incorporated in thermoreversible gels for age-related macular degeneration,” Pharm. Dev. Technol., 2016, doi: 10.3109/10837450.2014.965326.
[17] M. L. Bruschi, Ed., “5 - Mathematical models of drug release,” in Strategies to Modify the Drug Release from Pharmaceutical Systems, Woodhead Publishing, 2015, pp. 63–86.
[18] W. Liu, C. Lin, and Y. Hsieh, “Nanoformulation Development to Improve the Biopharmaceutical Properties of Fisetin Using Design of Experiment Approach,” 2021.
[19] P. Lakhani et al., “Optimization, stabilization, and characterization of amphotericin B loaded nanostructured lipid carriers for ocular drug delivery,” Int. J. Pharm., vol. 572, p. 118771, 2019, doi: https://doi.org/10.1016/j.ijpharm.2019.118771.
[20] X.-L. Yu and Y. He, “Application of Box-Behnken designs in parameters optimization of differential pulse anodic stripping voltammetry for lead(II) determination in two electrolytes,” Sci. Rep., vol. 7, no. 1, p. 2789, Jun. 2017, doi: 10.1038/s41598-017-03030-2.
[21] E. Swider, O. Koshkina, J. Tel, L. J. Cruz, I. J. M. De Vries, and M. Srinivas, “Acta Biomaterialia Customizing poly ( lactic- co -glycolic acid ) particles for biomedical applications,” Acta Biomater., vol. 73, pp. 38–51, 2018, doi: 10.1016/j.actbio.2018.04.006.
[22] R. de Sousa Victor, A. Marcelo da Cunha Santos, B. Viana de Sousa, G. de Araújo Neves, L. Navarro de Lima Santana, and R. Rodrigues Menezes, “A Review on Chitosan’s Uses as Biomaterial: Tissue Engineering, Drug Delivery Systems and Cancer Treatment,” Mater. (Basel, Switzerland), vol. 13, no. 21, p. 4995, Nov. 2020, doi: 10.3390/ma13214995.
Round 2
Reviewer 1 Report
After the corrections have been made, the manuscript may be accepted for publication.
Author Response
Thank you for your comments and suggestion, which helped to improvise the paper.
Reviewer 2 Report
Manuscript has been imoroved. Minor revision neede.
The addition of 'figure 9' does not add value, and it looks clutrred. I suggested adding lines in the earlier version (figure 8, now figure 10). I guess that will avoid clutter as well improve the understanding.
Author Response
Reviewer Comment: The addition of 'figure 9' does not add value, and it looks clutrred. I suggested adding lines in the earlier version (figure 8, now figure 10). I guess that will avoid clutter as well improve the understanding.
Author Response: Figure 9 was added as an overall comparison of the different formulations. The same information was present in figure 10 as individual graphs. As suggested, figure 9 was removed and the lines were added to figure 10 (now figure 9).